

# The T-Bird – A new aircraft-towed instrument platform to measure turbulence and aerosol properties close to the surface: Introduction to the aerosol measurement system.

Zsófia Jurányi[1], Christof Lüpkes[1], Frank Stratmann[2], Jörg Hartmann[1], Jonas Schaefer[2], Anna-Marie Jörss[1], Alexander Schulz[1], Bruno Wetzel[2], David Simon[2], Eduard Gebhard[1], Maximilian Stöhr[1], Paula Hofmann[1], Dirk Kalmbach[1], Sarah Grawe[2], and Andreas Herber[1]

[1]Alfred Wegener Institute, Helmholtz Centre for Polar and Marine Research, Bremerhaven, Germany
[2]Leibniz-Institute for Tropospheric Research, Leipzig, Germany

**Correspondence:** Zsófia Jurányi (zsofia.juranyi@awi.de)

**Abstract.** This study introduces the T-Bird, a novel aircraft-towed platform developed to measure turbulence and aerosol properties close to the surface, particularly over sea ice and open water in the polar regions. The T-Bird system, towed by the Alfred Wegener Institute's Polar aircraft, offers a unique capability to capture data from altitudes as low as ∼10 m while the aircraft operates at its lowest allowed operation altitude. This measurement platform allows for simultaneous data collection of

turbulence, aerosol, and other atmospheric parameters across multiple vertical layers. The T-Bird is equipped with specialized aerosol instrumentation to assess particle number concentration, number size distribution and absorption coefficient and to collect filter samples. It has been tested under Arctic conditions during the BACSAM (Boundary layer and Aerosol and Cloud Study in the Arctic, based on aircraft and T-Bird Measurements) campaign in October 2022. This paper provides technical details on the T-Bird's design, with special focus on the aerosol instrumentation, and its performance during Arctic flights

addressing measurement challenges in the lowermost atmosphere. The first measurements demonstrate its potential to enhance understanding of aerosol dynamics and boundary layer processes in remote environments.

## 1 Introduction

Aerosols play a crucial role in Earth's radiative energy balance and climate, directly by scattering and absorbing solar radiation, and indirectly by modifying cloud microphyisical properties (e.g. Szopa et al., 2021). The global warming due to the increase

of greenhouse gas concentrations is accelerated in the Arctic (Jeffries et al., 2013; AMAP, 2021) as the result of the action of a multitude of processes and feedback mechanisms summarized by the term of Arctic amplification (e.g. Wendisch et al., 2017). One of these interactions within the Arctic climate system is related to changes of aerosol concentrations (AMAP, 2021). The natural aerosol baseline in the Arctic is changing rapidly, accompanied by significant regional variations. The majority of Arctic aerosols originate from sources at lower latitudes traveling long distances through the atmosphere (Barrie et al., 1992).

However, the recently growing human influence in the Arctic accentuates the significance of local pollutant sources as well (Law et al., 2017). Altogether, the role of long-range transported or locally emitted aerosol particles in Arctic amplification,





particularly aerosol-cloud interaction, is complex and still not well understood (Mauritsen et al., 2011; Pithan and Mauritsen, 2014; Wendisch et al., 2017). In this regard, more measurements are required to improve our understanding of aerosol effects (e.g. Schmale et al., 2021).

Limited accessibility, harsh weather conditions, and the vastness of the Arctic complicate long-term and comprehensive aerosol measurements. Nevertheless, several ground-based research stations and monitoring sites have been established since the 1970s to collect continuous, long-term aerosol data. Some of the oldest stations are Barrow/Utqiaġvik (Bodhaine, 1989; Polissar et al., 1999), situated in Alaska, and Zeppelin in Ny-Ålesund, Svalbard (Platt et al., 2022). In the 1980s Alert station in Canada (Sirois and Barrie, 1999) and Villum Research Station in Greenland (Heidam et al., 1999; Nguyen et al., 2016) have
started their operations in the high Arctic. These ground-based stations provide valuable information on the seasonal variations and long-term trends of aerosol properties. However, these provide data limited to single geographic points on the ground.

  Ship-based aerosol measurements can cover wide ranges of oceanic, sea-ice and coastal regions, providing a more extensive and diverse sampling of the Arctic environment (e.g. Chang et al., 2011; Tjernström et al., 2014; Wendisch et al., 2019). The most comprehensive atmospheric measurement program conducted over the Arctic sea ice to date was part of the MOSAiC
(Multidisciplinary drifting Observatory for the Study of Arctic Climate) campaign, which provided year-around, continuous aerosol observations from the central Arctic (Shupe et al., 2022).

  In situ data on the vertical distribution of aerosols are rare compared to data collected at ground level. In the last two decades aircraft campaigns have allowed researchers to study the vertical distribution of aerosols, gaining insights into their transport, mixing, and interactions with clouds (e.g. Yamanouchi et al., 2005; Brock et al., 2011; Willis et al., 2019; Wendisch et al.,
2019; Jurányi et al., 2023; Wendisch et al., 2024). These fast-moving platforms can reach almost any remote location except the lowermost part of the atmosphere below 60-100 m above ground.

  The Arctic atmospheric boundary layer (ABL) is very shallow, often less than 100 m deep, due to strong surface inversions and stable atmospheric conditions (e.g. Vihma et al., 2014; Peng et al., 2023). Surface processes like water/ice/snow-atmosphere exchange of gases and aerosol particles have a significant impact on atmospheric composition in the Arctic ABL.
Therefore, the lowest layer of the Arctic atmosphere exhibits unique dynamics, chemistry, and sensitivity to surface exchange that can only be resolved through in-situ measurements within this critical layer height. Arctic tethered balloon (Ferrero et al., 2019; Cappelletti et al., 2022; Lata et al., 2023; Pohorsky et al., 2024), and unmanned aerial vehicle (UAV) (Bates et al., 2013; de Boer et al., 2018) measurements can cover these lowest altitudes inaccessible for aircraft, and provide high vertically resolved information on the occurring aerosol processes. However, these platforms can only be operated from specific locations
with sufficient infrastructure available. Tethered balloons are fixed to a single point, whereas the operation range is very limited for a UAV. Towed bodies offer a possibility to perform long-ranged measurements either using a helicopter (Siebert et al., 2006; Pätzold et al., 2023) or an aircraft (Frey et al., 2009) to tow the instrument platform and eventually perform simultaneous measurements (aircraft and towed body) as well.

  In order to be able to perform long-range turbulence and aerosol measurements in the lowermost layer of the polar atmo-
sphere and simultaneously at a second altitude, the towed-body system, T-Bird, has been developed. Here, the focus is on the technical description of the system (with special focus on the aerosol instrumentation) and of its first application during the





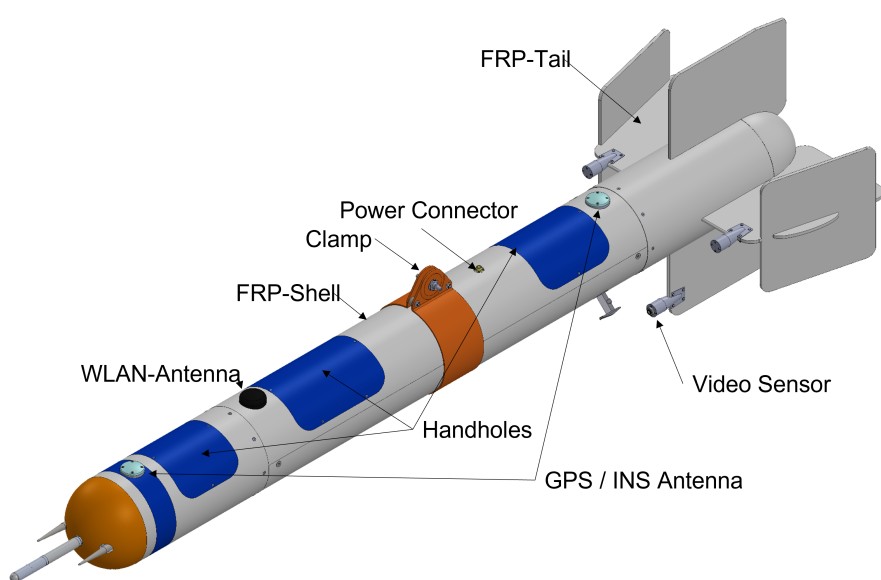

**Figure 1.** CAD-drawing of the T-Bird with main outer parts.

BACSAM campaign in the Arctic over the Fram Strait close to Svalbard. The technical description follows in the next section and a first case study is described in Section 3.3.1. There, we focus on the question whether the vertical aerosol concentration profile follows the ABL structure as obtained from potential temperature measurements.

## 2 The T-Bird–aircraft tandem measurement approach

### 2.1 Technical details of T-Bird

In this section we give a technical introduction to the T-Bird system. The T-Bird is a passive trailing body towed below the Alfred Wegener Institute's (AWI) polar aircraft Polar 5 or Polar 6. The body is attached to the aircraft by a cable of variable length, which also supplies power from the aircraft to the bird. It can be winched to a maximum of $\sim$100 m cable length allowing a vertical distance between bird and plane of about 60 m at typical measurement speed of 185-220 km h$^{-1}$. The actual rules of the air safety regulation allows aircraft measurements higher than 60 m above ground, the use of the T-Bird extends the measurement range to as low as $\sim$10 m above flat surface. This unique measurement setup allows us to measure key parameters (e.g. meteorological, turbulence, and aerosol properties) within the Arctic ABL at the important 10 m level with a simultaneous second measurement level above realized by the sensors onboard the airplane. Beyond this, various other dual-level-measurements in polar regions are feasible, that are desirable for e.g. cloud and aerosol physics studies.

The overall design of the T-Bird body, especially shape and size, is based on the successfully utilized EM-Bird (Haas et al., 2009). The outer body is made of fiber reinforced polymer (FRP), the inner structure of aluminium. The length of the T-Bird





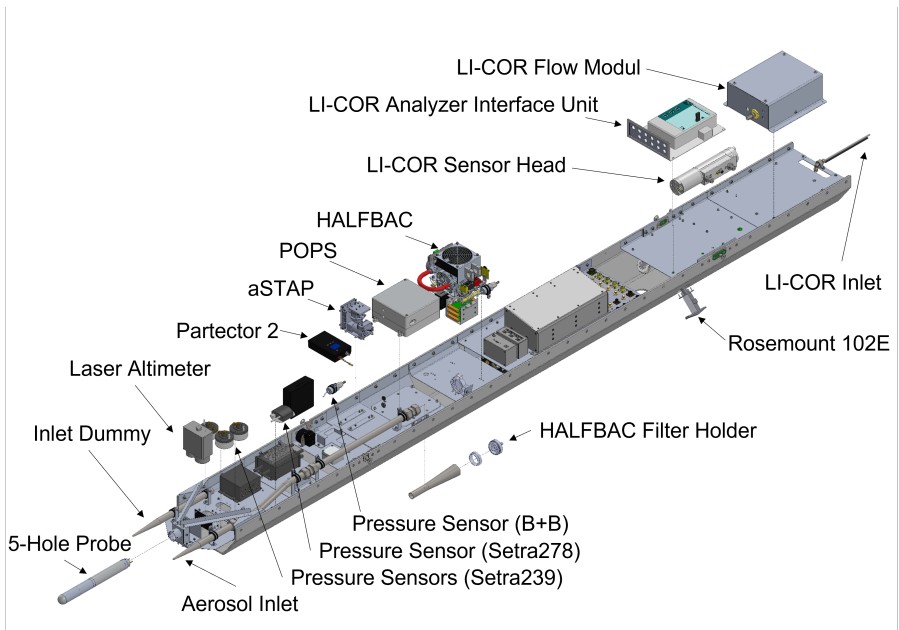

**Figure 2.** Exploding-view CAD-drawing of the T-Bird system including descriptions of main sensors parts.

including nose boom and tail is 396 cm, with a diameter of 35 cm. The stabilising tail has a width and height of 107 cm. The fully equipped system weighs about 108 kg. Figure 1 shows a CAD-drawing of the system. The T-Bird is anchored to the rope at a single clamp in the middle above its centre of gravity. The power connector is located behind the anchor clamp. Most data is sent in real time via wireless LAN to the data management system onboard the airplane. A video camera is located at the tail allowing real-time optical monitoring of the system environment during flight. Fully equipped, the installed instruments can be grouped into three payload types: aerosol, turbulence, and support equipment. Figure 1 shows where the different sensors are situated in the T-Bird, in the following sections the different types of sensors will be introduced, with special focus on the aerosol instrumentation.

### 2.1.1 Turbulence instrumentation

The turbulence equipment is in principal the same as mounted on the aircraft where it has been used successfully in the polar ABL during several campaigns (e.g. Ehrlich et al., 2019; Gryanik and Hartmann, 2022; Chechin et al., 2023). Namely, the same five-hole probe (Aventech) is installed at the front of the T-Bird (Fig. 2) and provides static and dynamic (Pitot) pressure and differential pressure measurements to derive the angles of attack and sideslip and finally the 3D wind vector with a frequency of 100 Hz (Hartmann et al., 2018). A LI-COR infrared gas analyser (LI-COR Biosciences, LI-7200) measures gas-phase $H_2O$ and $CO_2$ concentrations through an actively aspirated and heated gas inlet situated at the back of the system. The turbulence instrument bunch is completed by a Pt100 temperature sensor in a Rosemount housing (Rosemount, type 102E) mounted at the underside of the T-Bird's rear third. The turbulence payload allows the determination of the momentum, sensible and latent





heat fluxes, and as will be shown in a later research paper (Simon et al., in preparation), in combination with high frequency
measurements of aerosol particle concentrations, also particle fluxes.

### 2.1.2 Support instrumentation

Position, heading and altitude information of the T-Bird is provided by a GPS supported inertial navigation system (INS, IMAR
Navigation & Control iNAT-M200-SLN-DA) during flight. These are essential measurements to correct the wind vector from
the five-hole probe for the T-Bird's own movements. A camera system (AXIS F1005-E sensor and AXIS F41 main unit) was
installed with the camera sensor localized at the T-Bird's tail looking in flight direction to document the overflown underground.
Furthermore, a laser altimeter (ASTECH LDM 301) measures the height of the T-Bird above ground.

### 2.1.3 Aerosol instrumentation

The T-Bird's aerosol inlet (Enviscope/Knebel, see Fig. 1) is mounted at the nose, next to the five-hole probe but behind it to
minimise its influence on turbulence measurements. In order to keep the T-Bird's symmetry and with that its balance, a blind
copy of the inlet was placed on the opposite side of the five-hole probe. The inlet is constructed of stainless steel, the ambient
air first enters a cone shaped diffuser (131.7 mm length, 3.7 mm opening diameter and $4\,°$ opening angle) to decelerate it. This
enables isokinetic sampling at a total flow of $3.2\,\mathrm{l\,min^{-1}}$ and flight speed of $60\,\mathrm{m\,s^{-1}}$. The inlet enters the T-Bird through a
22.1 mm diameter stainless steel tube, the individual sample lines to the various instruments were drawn from that tube using
angled inserts and connected to the individual instruments by conductive rubber tubing. Or, in case of the Partector instrument,
we applied PVC (Tygon) tubing, as recommended by the manufacturer (Asbach et al., 2016).

The amount of heat produced by the instruments during operation was high enough during the Arctic autumn test campaign
BACSAM to keep them within their operational temperature range. The higher inside temperature of the T-Bird was also
enough to keep the relative humidity of the aerosol sample below 40%, no additional drying was used. An optional heating
system can be installed for campaigns performed at even lower temperatures present.

Due to the limited space and payload capacity of the T-Bird, only miniaturized aerosol instruments could be installed in
it: a Partector 2, a portable optical particle spectrometer (POPS), a single channel tricolor absorption photometer (STAP), and
a high-volume and light-weight balloon-borne filter sampler (HALFBAC). Table 1 shows a list of the T-Bird aerosol instru-
mentation, the measured properties and their time resolution. In the following we give a short introduction to the individual
instruments.

A partector 2 (Naneos particle solutions GmbH, Windisch, Switzerland) was installed in the T-Bird to measure the particle
number concentration and the average diameter of particles in the diameter range of ∼10–300 nm. The instrument's measure-
ment principle is based on unipolar corona diffusion charging and subsequent electrical current measurement. The unipolar
charger is operated in a pulsed mode, which leads to packets of charged aerosol inducing currents in a two-stage Faraday Cage
connected to high-sensitivity electrometers. Between the two Faraday cages, an electrostatic precipitator is installed, where part
of the charged aerosol particles are removed (Fierz et al., 2014). The measured two currents are dependent on the particles'
size, and concentration. Using assumptions on the form of the particle number size distribution (having lognormal form with a





certain width and being monomodal), the two-stage measurement makes it possible to determine both, the total aerosol particle number concentration and an average particle diameter. The instrument's miniature size ($8.8 \times 14.2 \times 3.4$ cm), light weight

(415 g) and the lack of the need of a condensation fluid (butanol or water, that is needed by the most commonly used particle counters) makes it an ideal instrument to be operated in the T-Bird.

The POPS (Handix scientific, Fort Collins, CO,USA) is an optical particle counter and was developed for UAV and balloon deployment (Gao et al., 2016; Mei et al., 2020). Its small size and low power consumption makes it ideal for our applications. The instrument operates at a wavelength of 405 nm and optically sizes the sampled particles based on their single-particle

elastic light scattering in the diameter range approximately 140 nm to 3 µm. The POPS works with a sample-sheath flow system, where a single miniature pump is responsible for establishing both flows simultaneously. The pump draws air through the optical chamber from the instrument's aerosol inlet and the same time ambient air through an orifice and a particle filter as the sheath air. The sample flow rate through a laminar flow element is measured by a differential pressure sensor, and the pump speed is controlled such that the sample flow stays constant at $200 \, \text{cm}^3 \, \text{min}^{-1}$. The particle free sheath air ensures that

the particles pass through the middle of the instruments laser beam. It is not monitored in the system, but the orifice (normally) guarantees its stable flow of $150 \, \text{cm}^3 \, \text{min}^{-1}$.

The STAP (Brechtel Inc, Hayward, CA, USA – Model 9406) was built based on the Particle Soot Absorption Photometer (PSAP, Bond et al., 1999) instrument. Diffused light from three LED sources centered around wavelengths of 445, 515, and 633 nm (blue, green, red) are alternatingly transmitted through two glass fiber filters, the transmitted light is monitored by

photodetectors. One of the filters serves as a reference and no particles will be sampled through, the other will be loaded with aerosol particles. Decrease in the transmitted light indicates the accumulation of absorbing particles on the filter, and the light absorption coefficient at the three wavelengths are reported. These absorption coefficients can be converted to equivalent BC (eBC, Petzold et al., 2013) concentrations. The instrument's lower detection limit is defined by its noise level, which is $0.2 \, \text{Mm}^{-1}$ at 60 s averaging time (Bates et al., 2013). This translates to 10–20 ng m$^{-3}$ eBC dependent on the applied mass

absorption efficiency.

The filter sampler used for the collection of aerosol particles in the T-Bird is based on the TROPOS-built High-volume And Light-weight Filter samplers for BAlloon-borne appliCation (HALFBAC, Grawe et al., 2023). It features a filter holder (47 mm, in. inlet; PFA, Savillex, MN, USA); a vacuum scroll pump (SVF-E0-50PF, Scroll Labs, USA); as well as temperature, pressure before and behind the filter, volume and mass flow sensors. The instrument is controlled by means of a Raspberrry PI, which is

utilized for collecting the measured data and pump control. 800 nm pore size polycarboante filters (Nuclepore™ track-etched membranes, Whatman, UK) were used, the sample flow rate was set to $30 \, \text{l} \, \text{min}^{-1}$, resulting in sampled air volumes per filter between 2 and 7.2 m$^3$.

## 2.2 Polar 6 aircraft and BACSAM aircraft campaign

Polar 6 is one of the research aircraft of Alfred Wegener Institute and was deployed to tow the T-Bird, and to host measurement

instrumentation onboard as well. The aircraft is a specifically modified Basler BT-67 aircraft for polar missions and has the





ability to fly at low cruising speeds of 185–400 km h$^{-1}$ performing measurements in an altitude range from 60 to 8000 m (Wesche et al., 2016).

The Boundary layer and Aerosol and Cloud Study in the Arctic, based on aircraft and T-Bird Measurements (BACSAM) took place between 1 October 2022 and 16 October 2022 over the Fram Strait with an operation base of Longyearbyen airport

(78°14'43" N, 15°28'57" E) in Svalbard. The aircraft campaign was carried out within the framework of the project Arctic Amplification: Climate Relevant Atmospheric and Surface Processes and Feedback Mechanisms ((AC)[3], Wendisch et al., 2023).

The main goal of the campaign was to test the T-Bird for the first time in Arctic conditions, including dynamic flight behaviour characterisation, T-Bird's instrumentation calibrations, testing and comparison, and last but not least performing

aerosol and turbulence measurements with both the T-Bird and the aircraft across the ABL up to the FT over open ocean and over the marginal sea-ice zone. All together nine scientific flights were carried out with a total 29 hours of flight time.

## 2.3 Aircraft turbulence instrumentation

As already mentioned, the aircraft was equipped with a nose boom hosting the instrumentation bunch for turbulence observations. This consist of a five-hole probe (Aventech) and Pt100 temperature sensor (Rosemount, type 102E). Additionally, there

are a deiced (heated) Pt100-temperature sensor (Rosemount, type 102E) and a humidity sensor (Vaisala HMT333) mounted on Polar 6 with dedicated inlets. Furthermore, a LI-COR Biosciences, LI-7200 system for H$_2$O and CO$_2$ concentration measurements is installed in the aircraft cabine with an inlet on the aircraft's roof.

## 2.4 Aircraft aerosol instrumentation

The aircraft aerosol inlet with an intake diameter of 0.35 cm is located ahead of the engines, all aircraft aerosol instruments

sampled air through this shrouded stainless-steel inlet diffuser. At the typical cruising speeds of Polar 6 the inlet has a close to unity transmission efficiency in the particle diameter range of 20 nm to 1 μm (Leaitch et al., 2016). The relative humidity of the aerosol sample was always below 30%, due to the higher cabin temperature compared to the ambient temperature, no additional measure was needed to dry the aerosol.

The Polar 6 aerosol instrumentation, installed in the aircraft cabin during BACSAM consisted of a single particle soot

photometer (SP2), a scanning mobility particle sizer (SMPS) and a high-volume flow aerosol particle filter sampler (HERA).

The SP2 (Droplet Measurement Technologies, Longmont, CO, USA) is able to measure BC mass of individual aerosol particles in the mass equivalent diameter range from ∼80 nm to ∼600 nm (assuming void-free bulk material density of 1.8 g cm$^{-3}$ (Moteki and Kondo, 2010)). The BC detection is based on laser-induced incandescence, whereby a continuous-wave, high-intensity, intra-cavity laser (Nd: YAG crystal, wavelength of 1060 nm) heats up absorbing particles until they reach their

vaporization temperature and emit incandescent light. Its intensity is proportional to the BC mass of the particle (Schwarz et al., 2006; Moteki and Kondo, 2010). The SP2 was calibrated before and after the measurement campaign using size selected fullerene particles (Moteki and Kondo, 2010; Gysel et al., 2011; Laborde et al., 2012). The calibration curves before and after



the campaign agreed within 5%, and therefore we can assume that the instrument's sensitivity remained constant during the campaign.

The SMPS (Wiedensohler et al., 2012) is a custom-built system, which consists of a differential mobility analyser (Vienna-type, custom built) and a condensational particle counter (TSI CPC3760A; TSI Incorporated, USA). It measures the aerosol number size distribution within the particle mobility diameter range of 10–850 nm, was operated with a sheath flow of $5\,l\,min^{-1}$ and sample flow of $1\,l\,min^{-1}$. A single SMPS scan lasted $300\,s$.

For the sampling of aerosol particles for subsequent offline analysis, e.g. concerning INP abundance and properties, the

TROPOS-developed HERA sampler was used. HERA is an aerosol filter sampler for airbone-applications which has been described in detail in Grawe et al. (2023). It features a revolver-like-arranged set of six filter holders, through which the sample flow is guided by a ball valve. During the T-Bird test campaign, the volumetric sample flow rate of HERA was set to $30\,l\,min^{-1}$. As filters, 800 nm pore-size polycarbonate filters (Nuclepore™ track-etched membranes, Whatman, UK) were used. The sampling time varied between 16 and 1666 min, resulting in sampled air-volumes between 0.4 and $4.5\,m^3$.

| Platform | | Instrument | Measured property | Size Range | Time resolution |
|---|---|---|---|---|---|
| Polar 6 | Aerosol | SMPS | Number size distribution | 10–850 nm | 300 s |
| | | SP2 | BC mass concentration, size distribution | 80–600 nm | 1 Hz |
| | | HERA | Filter Sampling | - | 6 filters per flight |
| Polar 6 | Turbulence | 5-hole probe | 3-d wind components | - | 100 Hz |
| | | gas analyzer | $CO_2/H_2O$ concentration | - | 20 Hz |
| | | Pt100 | total air temperature | - | 100 Hz |
| T-Bird | Aerosol | Partector | Number concentration, average diameter | 10–300 nm | 1 Hz |
| | | POPS | Number size distribution | 140–3300 nm | 1 Hz |
| | | aSTAP | BC absorption coefficient | total | 0.5 Hz |
| | | HALFBAC | Filter Sampling | - | 1 filter per flight |
| T-Bird | Turbulence | 5-hole probe | 3-d wind components | - | 100 Hz |
| | | gas analyzer | $CO_2/H_2O$ concentration | - | 20 Hz |
| | | Pt100 | total air temperature | - | 100 Hz |
| T-Bird | Support | INS | 3d angles | - | 100 Hz |
| | | altimeter | height (agl) | - | 100 Hz |
| | | camera | color images | - | 50 Hz |

**Table 1.** List of the scientific instrumentation including the measurement platform, the description of the measured quantity, size range and time resolution.





## 3 Results

### 3.1 T-Bird's flight behaviour

To make a statement about the flight characteristics of the T-Bird, a special flight test was carried out during the BACSAM campaign. In this test flight, a series of previously defined manoeuvres were flown to provide information on the interactions between the T-Bird and the towing aircraft. The manoeuvres were based on the explanations of Jategaonkar (2006) and were used to analyse the longitudinal and lateral motions of the T-Bird. Using INS and GPS data, a state space model was created. This revealed excellent flight characteristics of the T-Bird. Therefore, the T-Bird is characterised by uncritical flight behaviour with good damping behaviour in lateral motion, which makes it ideal for the scientific mission. More detailed investigation could be made in further flight tests and through force measurements at certain angles of attack using wind tunnel tests.

### 3.2 Instrument performance in the T-Bird and in the aircraft

During BACSAM, most of the aerosol instrumentation functioned properly both in the aircraft and in the T-Bird as well, including the SMPS, HERA and SP2 in Polar 6 and Partector 2, HALFBAC in the T-Bird. We have encountered some issues with the POPS, as described below, however this problem will have been solved in future applications (Simon et al., in preparation).

The aSTAP showed no indication of malfunctioning, however during the BACSAM aircraft campaign we encountered extremely low BC concentrations below $1 \, \mathrm{ng \, m^{-3}}$. This concentration is even much lower than the Arctic average summer season (out of Arctic haze season) concentration of $4.7 \, \mathrm{ng \, m^{-3}}$ (Jurányi et al., 2023), and with that, the BC concentration was well below the detection limit of the instrument. The only times when the instrument delivered values above the noise level was directly before take off and after landing, when the aircraft emissions were sampled. With that, during BACSAM we only could test that the instrument is functional and that the data acquisition works properly. During future deployments, in spring season and/or closer to local sources we expect much higher BC concentrations above the instrument's detection limit, and with that valid measurements.

The POPS encountered flow problems during BACSAM. The instrument works with a sample-sheath flow system, where the particle-free sheath flow ensures that the aerosol particles pass through the middle of the laser beam. With the ram pressure present at the instrument's inlet during flight, the sample flow was always higher than the desired $200 \, \mathrm{cm^3 min^{-1}}$. As a consequence the instrument's pump was regulated down completely, the sheath flow was below the desired value, or even reversed. Additionally, the sample flow was as high that it could not be measured anymore (higher than the highest value that the laminar flow element could detect). After realizing this problem, as a fast solution, the inlet tube of the instrument was perforated such that the ram pressure induced excess flow was diverted before entering the instrument. This makeshift solution did not completely solve the problem, but at least the sample flow remained in the measurable range during the slower flight sections when the T-Bird was winched out. In following campaigns this problem has been avoided by placing a bypass tubing between the inlet and outlet of POPS.





The filter sampling instruments HALFBAC and HERA worked properly during the BACSAM campaing, 8 Filter samples with HERA and 18 with HALFBAC could be collected for INP analysis. The filter analysis is still in progress, a comparison between HALFBAC and HERA collected filter results will be subject of a separate future publication.

The main goal of the T-Bird application during future campaigns will be to obtain the vertical aerosol distribution in the
entire ABL focusing on its lowest layers down to 10 m above the surface. This will be done with simultaneous measurements at two altitudes such that highly resolved vertical profiles can be obtained with only few horizontal flight sections in different heights. Such missions involving flights with the T-Bird at the lowest possible altitude above ground were not within the scope of the first test campaign, the flight behaviour of the system was rather tested at safer, higher altitudes. Therefore, most of the time during the scientific flights, it can be expected, that both aircraft and T-Bird sampled aerosols from the same atmospheric
layer with only negligible vertical gradients of aerosol properties. This gives us the opportunity to investigate the performance of the „miniature" aerosol instrumentation deployed in the T-Bird, by comparing their measured quantities to the „standard" instruments onboard of the Polar 6 aircraft. Though, we still have to keep in mind the above described instrumental problems.

### 3.2.1   Comparison of Partector and SMPS Results

The aerosol number concentration data from the partector instrument deployed in the T-Bird was compared to the number
concentration derived from the aircraft's SMPS measurements. All concentration data in this manuscript is reported as ambient concentration and is not corrected to standard conditions. For this comparison all flight data was considered excluding solely take-off and landing. The data was separated according to the position of the T-Bird compared to the aircraft: i.e. we distinguish data when the T-Bird was in the nest directly below Polar 6 or when it was winched out completely ($\sim$60 m altitude difference between T-Bird and aircraft). With the T-Bird in the nest, the inlets of both systems are only separated by some meters and with
that, sample almost the same air. For this situation we expect both instruments' delivered aerosol number concentration values to agree to a great extent, and the arising differences give us information on the instrumental and measurement uncertainties. For the case with winched out T-Bird, some additional uncertainty might originate from sampling different aerosol, but as it was mentioned before, due to the chosen flight patterns, we expect that only a small fraction of measurements are taken in layers with strong vertical gradients of aerosol properties. Anyhow, the winched out T-Bird case was treated separately to
check the extent of such an influence.

The measured SMPS number size distributions were integrated in the diameter range of 10–300 nm in order to match the size range of the partector. The 1-second data from the partector was averaged to the 5-minute scan interval of the SMPS. Here, we have to take into account, that the SMPS counts the aerosol particles within a certain narrow diameter interval and scans through the considered diameter range. Therefore, when the aerosol concentration highly fluctuates within the scan time of the
SMPS, the derived aerosol number concentration will be erroneous. Thus, such cases were sorted out after a manual inspection of each individual SMPS up- and down-scans.

The comparison of the aerosol number concentration between the partector ($N_{\mathrm{p}}$) and the SMPS ($N_{10-300}$) is shown in Figure 3 for both cases of winched in (green round markers) and winched out (red triangles) T-Bird. First of all, it is obvious that partector data and SMPS data agree well with each other independent on the case confirming the assumption that most



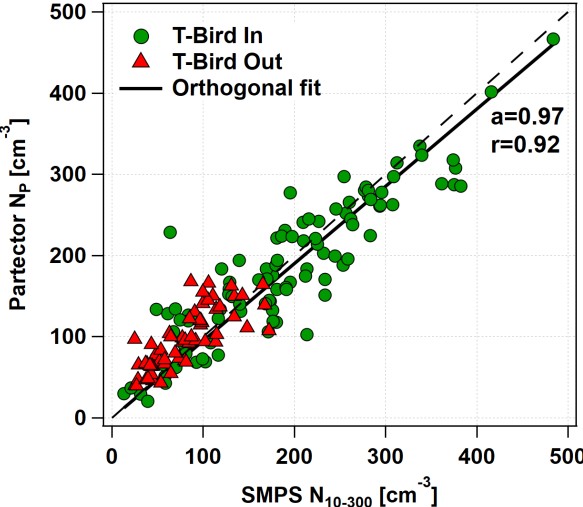

**Figure 3.** Comparison of the measured aerosol number concentration between the partector 2 and the SMPS between 10 and 300 nm particle diameter during the BACSAM campaign. The green round markers represent the periods when the T-Bird was directly below the aircraft, the red triangles when the T-Bird was winched out. The black solid line is the corresponding orthogonal regression line, the black dashed line shows the line of equality to guide the eyes.

of the time we sampled the same aerosol. The only difference between the winched-in and winched-out case is the smaller concentration range of data for the latter case. The reason is that the T-Bird was only winched out during flying closer to the surface (below ∼1000 m altitude), whereas transit flights (with T-Bird in its nest) took place at higher altitudes (between 2000-3000 m), where significantly higher concentrations were encountered during BACSAM.

Based on this, we have chosen to investigate the relationship between the measured number concentration from the two

different instruments without splitting the data according to the T-Bird's position. The orthogonal distance regression line (green line) forced through the origin shows that on average the Partector reported an $N_p$ close to the SMPS with a slope of 0.97, and the data is highly correlated (correlation coefficient of 0.92). Despite the very good agreement some individual measurement points can still show high scatter. It also has to be mentioned here, that due to the working principle of the partector, the original 1-second time resolution data (with a 4-second integration time setting) shows very high noise at such

low concentrations. The average ratio between $N_p$'s standard deviation and mean value was 1.09 which means that the level of noise is comparable to the level of the measured signal. This should be considered for the further data analysis.

The average diameter obtained by the partector can also be compared to the measurements of the SMPS, and for this comparison the geometric mean diameter of the size distribution was chosen. Even with perfect instrument performance, we do not expect an agreement between these values, since the „average diameter" reported by the partector is based on many

assumptions as mentioned in section 2.1.3. Only if the measured number size distribution exactly fulfilled all these assumptions, the partector's average diameter would agree with the geometric mean diameter of the number size distribution. The partector





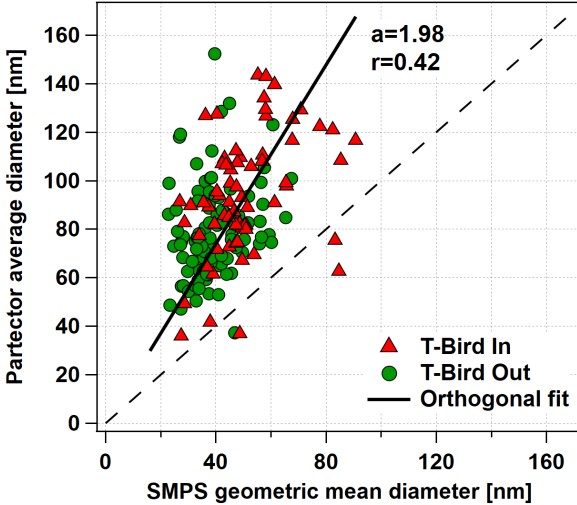

**Figure 4.** Comparison of the reported average diameter by the Partector 2 to the number mean diameter from the SMPS. The green round markers represent the periods when the T-Bird was directly below the aircraft, the red triangles when the T-Bird was winched out. The black solid line is the corresponding orthogonal regression line, the black dashed line shows the line of equality to guide the eye.

data was again averaged through the duration of the single SMPS scans, and the same criterion for not considering the points with too highly fluctuating aerosol concentration was applied as before. The comparison is presented in Figure 4, showing the cases according to the T-Birds position separately (green round markers vs. red triangles).

The partector reports on average almost a factor 2 (orthogonal regression fit line has a slope of 1.98) higher diameter than the geometric mean diameter measured by the SMPS, with a correlation coefficient of 0.42. The spread of the individual measurement points is high but no difference between the two cases regarding the T-Bird's position can be identified. Based on this, we can conclude that the average diameter from the partector cannot be directly compared to the geometric mean diameter of the number size distribution. Nevertheless, results might still be used as an indicator of the average size of the ultrafine

particle size range.

### 3.2.2 Comparison of POPS and SMPS Results

The data originating from the POPS in the T-Bird was compared to the Polar 6's SMPS system. The instruments have a sufficient overlap within their size ranges (SMPS: 10–850 nm, POPS: 153–3000 nm) from 153 to 850 nm. To elucidate the effects of the above described flow problems of the POPS, we carried out a comparison of the SMPS and POPS measured size distributions.

Thereto, the data was sorted according to sample flow being within the instruments measurement range (flow in range) or being even higher (flow out of range). The number size distributions were integrated between 153 and 850 nm for both instruments and the 1-second POPS data was averaged for the duration of the SMPS scan. SMPS scans were again excluded from the comparison, for which the POPS concentration fluctuated too much within the time period of one SMPS scan.



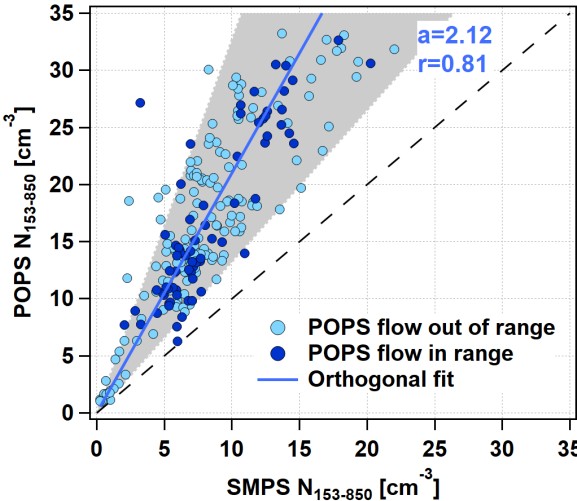

**Figure 5.** Comparison of the measured aerosol number concentration between the POPS instrument and the SMPS between 153 and 850 nm particle diameter during the BACSAM campaign. The dark blue markers represent the periods when the POPS sample flow was in the measurement range, the light blue markers the ones when the flow was even higher. The blue line is the corresponding orthogonal regression line, the black dashed line shows the line of equality to guide the eye. The light grey shaded area ends at the 95[th] and 5[th] percentiles of the ratio between the POPS and SMPS concentrations.

The comparison of the number concentration is shown in Figure 5: dark blue markers correspond to periods with a mea-
surable sample flow, the light blue markers show the observations when the sample flow exceeded the maximum range of values that can be measured by the instrument. Firstly, there is no significant difference between the two cases, secondly, the POPS number concentration is highly correlated (correlation coefficient of 0.81) to the measured SMPS number concentration. However, due to the flow problem, it seems that the POPS on average counted ∼2 times higher concentration than the SMPS (slope of the orthogonal fit is 2.12). 90% of the points can be found within 1.22 and 3.26 times the number concentration of
the SMPS. This area is shown as a grey shading in Figure 5. This factor of 2.12 can be considered as the „counting efficiency" of the instrument.

To be able to investigate solely the sizing performance of the instrument, the POPS number size distributions were corrected with the previously determined „counting efficiency" factor. The median number size distributions after this correction are presented in Figure 6a for the case when the POPS sample flow was out of the measurement range, and in Figure 6b for the case
when the sample flow could be measured. For both flow cases the median number size distributions are not too far from each other and the ranges between 25[th] and 75[th] percentiles well overlap. Generally, the POPS seems to overestimate the number of the „smaller" (<250 nm) particles and underestimate the number of the „larger" (>250 nm) particles. This difference can partly exist because we compare different types of number size distributions with each other. The SMPS measures the number size distribution based on the mobility diameter whereas the POPS measures based on the optical diameter using Polystyrene
Latex particles with a refractive index of $1.615 + 0.001i$ (Gao et al., 2016) for calibration. However, we expect that this effect is



negligible compared to the sizing uncertainty caused by the mis-adjusted sheath flow. The too low, completely missing or even reversed sheath flow has the consequence that aerosol particles that were supposed to pass through the middle of the laser beam might have passed the laser closer to the edge of the beam with significantly lower intensity and therefore falsely identified as smaller particles. In the measured number size distribution, this would appear as measuring too many smaller particles and too

few larger ones, just like we have observed in our case.

As the measured number concentration highly correlates to the number concentration of the SMPS, we will still use the POPS data in the following as an indicator for the number concentration of larger (>153 nm) particles.

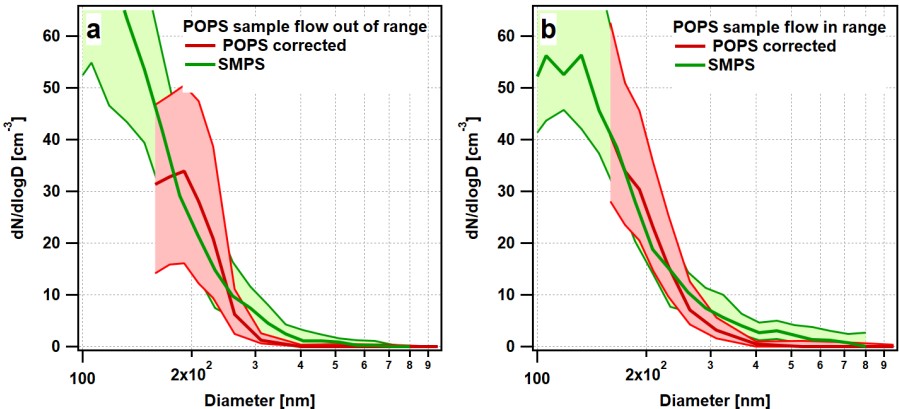

**Figure 6.** Comparison of the median number size distributions in the size range of 100 to 900 nm between POPS (red line, corrected to match the SMPS number concentration) and SMPS (green line) for the periods when the POPS sample flow was out of the measurement range (a) and when it was within the measurement range (b). The shadings were drawn between the 25$^{th}$ and 75$^{th}$ percentiles.

### 3.3   Case Study

In the following, we will focus on one flight mission as an example to show the influence of mixing processes in the lower

atmosphere (here below 2500 m) on the vertical distribution of aerosol particles. This flight took place on 05 October 2022 with the following flight pattern: after take-off, Polar 6 flew to the target area, followed by horizontal flight sections at increasing altitudes between two waypoints (WP1 and WP2). After that, the aircraft returned to the airbase and landed (Fig. 7).

#### 3.3.1   Research flight on the 05.10.2022

05 October 2022 was the date for the first scientific research flight. Due to the pronounced lee effect caused by Svalbard's

orography and prevailing easterly wind, a cloud-free area was formed west of Svalbard, which became the target area of the flight mission. Within this area, a northern WP1 and a southern WP2 were defined (Fig. 7), and between them, horizontal legs of approximately 10 minutes duration (equivalent to about 65 km distance) were performed at eight different altitudes (Fig. 8) above the open water surface. The T-Bird was winched out shortly before the first horizontal leg and was winched into its nest



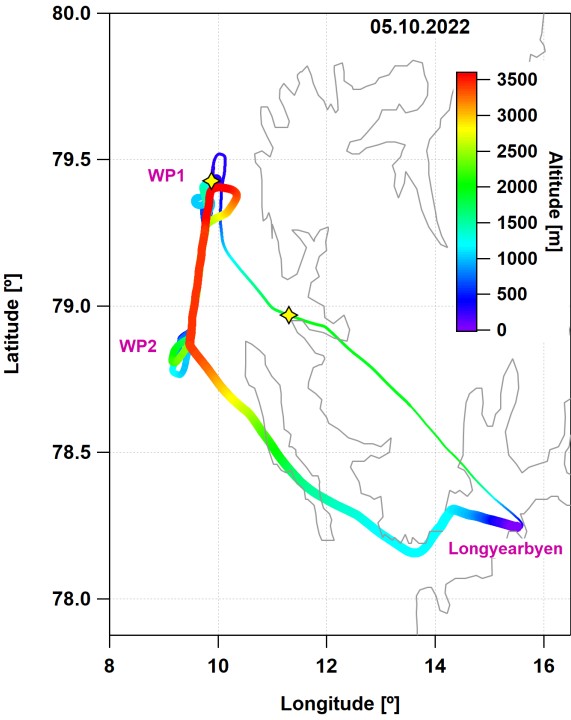

**Figure 7.** Flight pattern on the 05 October 2022, the different colors indicate the flight level of the aircraft, the thickness of the pattern increases with the elapsed time. The yellow stars show the start and end points of the profile measurement.

at the end of the sixth leg. Therefore, during the measurements of the two legs with the highest altitudes, the T-Bird measured at the same altitude as the aircraft (Figure 8).

Before reaching WP1 for the first time, the aircraft descended to the target area, and this provided us vertical profiles of meteorological quantities to investigate the structure of the lower atmosphere. Note that the descent started already close to the western tip of Svalbard (Figure 7, yellow star) and one should be aware of the fact that for this reason the potential temperature profile includes to some extent the effect of horizontal inhomogeneity along the distance during descent. Namely, below 400 m, e.g. the potential temperature changed in the order of 1–2 K between WP1 and WP2 (not shown). Thus, most accurately the

stability in the lowermost layer can be analysed from leg averages. These leg averages are shown in Figure 9, whereas Figure 10 (grey solid line) shows the potential temperature as a function of the altitude during the descent to WP1.

The temperature profile reveals that the structure of the lower atmosphere is characterized by multiple inversions and mixed layers in between starting with the near-surface ABL capped by a strong inversion at about 100-130 m height. Above the

inversion a well mixed layer (in the following called residual layer) follows again. And also this layer is capped by a strong inversion at about 750 m height, which reaches to about 1000 m. Then, another layer follows, which is also mixed but not so well as the residual layer. Finally, another inversion starts at about 1500 m height.





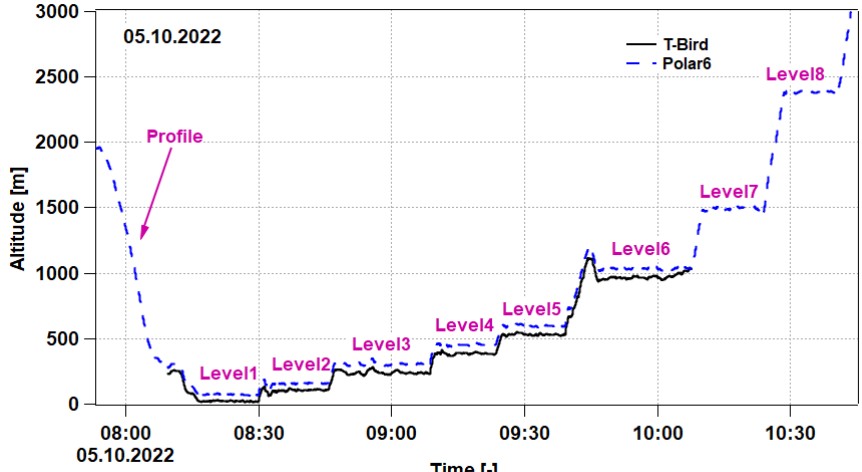

**Figure 8.** Flight altitude of the Polar 6 aircraft (blue dashed line) and the T-Bird (black solid line) as function of time on the 05 October 2022, measured by GPS. The T-Bird had only GPS reception when the system was winched out.

The leg averages show the potential temperature, wind speed (Figure 9 a, b) and turbulence quantities, namely sensible heat flux and turbulent kinetic energy (TKE) (Figure 9 c, d). For the calculation of these turbulence quantities, the linear trends of

wind and temperature between both way points have been eliminated. Both the profiles of potential temperature and heat flux point to a weak convectively mixed ABL below about 100 m height with small upward fluxes of sensible heat. It can be seen that the results from T-Bird (purple markers) for potential temperature and wind showing a weak low-level jet, as well as fluxes of sensible heat and turbulent kinetic energy fit very well to the results obtained from Polar 6 (green markers). This is remarkable, especially for the very small heat flux with values close to the detection limit. It shows a maximum near the surface

and - as it should be in a convective ABL - decreasing values below the capping inversion to near zero in the inversion. Also the TKE altitude dependence with a maximum near the surface is physically reasonable for a shear driven ABL.

In order to investigate if and how the aerosol properties vary within these above mentioned different atmospheric layers, median values and the $25^{th}$ and $75^{th}$ percentiles of the measured particle number concentrations encountered during each horizontal leg were determined and are marked by horizontal whiskers in Figure 10. Since the SMPS instrument has a 5-

minutes time resolution, during one altitude leg only 1-3 full scans could be performed. These scans were averaged and the number size distribution was integrated between 10 and 300 nm to obtain a number concentration comparable to the partector. The measured aerosol concentrations as a function of the measurement altitude are shown in Figure 10.

The median $N_{\mathrm{p}}$ measured by the partector stays relative stable with increasing altitude through the boundary and residual layers ($\approx$20–550 m) with values between 63 and 101 cm$^{-3}$. Such low number concentrations are not unusual outside of the

Arctic haze season in the region (e.g. Heintzenberg et al., 1991; Kupiszewski et al., 2013; Freud et al., 2017). Above the residual layer $N_{\mathrm{p}}$ increases with the altitude in the free troposphere (FT) reaching a median concentration of $\approx$300 cm$^{-3}$ at 2400 m altitude. $N_{10-300}$ measured by the SMPS features similar behaviour as $N_{\mathrm{p}}$, with constant concentrations within the boundary





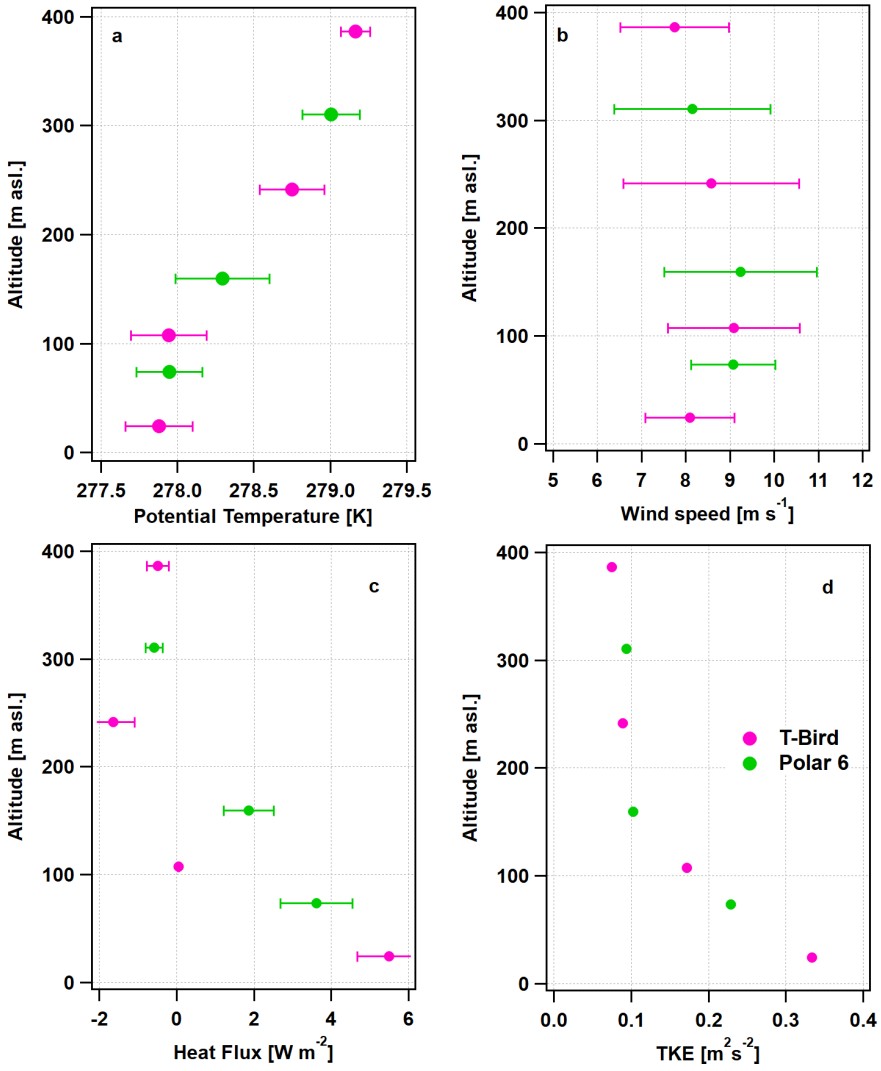

**Figure 9.** Turbulence measurements as function of altitude during the research flight on 05 October 2022 measured by Polar 6 (green markers) and the T-Bird (purple markers). Panel a shows the potential temperature (standard deviation as error bars), Panel b the wind speed (standard deviation as error bars), Panel c the sensible heat flux (sampling error as error bars) and Panel d the turbulent kinetic energy.

and residual layers with concentrations of 91–101 $cm^{-3}$, and increasing concentration within the FT and a highest value of 386 $cm^{-3}$. It is remarkable that the strong capping inversion above 100 m height has only little influence on the concentrations

while there are changes in concentrations at the inversions in the higher layers.

The situation changes if the number concentration of larger particles is investigated. The median of the corrected POPS number concentration between 153 nm and 3 µm, is shown as purple dots in Figure 10 corresponding to the purple x-axis. Due to the instrumental problems, we take these values as half-quantitative measure for the amount of large particles. Thanks to the



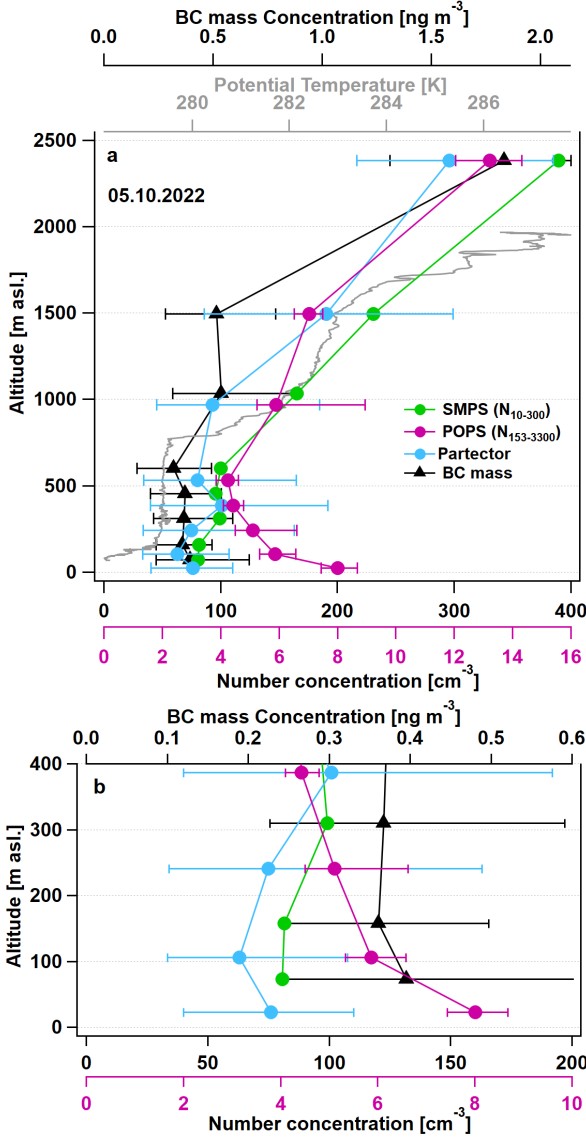

**Figure 10.** Aerosol properties as function of the altitude on 05 October 2022. The round markers show median aerosol number concentrations (blue: partector 2, green: SMPS integrated between 10 and 300 nm and purple: POPS corrected number concentration, bottom purple x-axis), the black triangles (top black axis) show the median BC mass concentration whereas the solid grey line shows the potential temperature (top grey axis). The error bars show the $25^{th}$ and $75^{th}$ percentiles. Panel a shows the entire altitude range, whereas panel b shows the same data zoomed into the lowermost altitude range up to 400 m altitude for better visibility.

technology of the T-Bird, we were able to perform measurements as low as $\approx 20$ m and this reveals a decreasing concentration

from the ABL to the residual layer, which points to a source of large particles at the surface, which is probably sea-spray. Apart from this, the behaviour of the larger particles at higher altitudes follows the behaviour of the number concentration of the





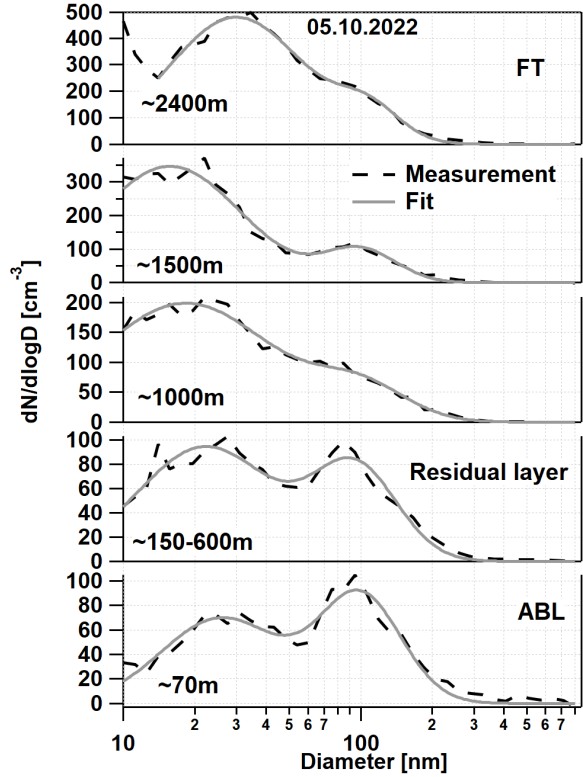

**Figure 11.** The averaged aerosol number size distribution at different atmospheric layers during the research flight on 05 October 2022. The black dashed lines show the measurement, the solid grey lines the fitted double lognormal distributions.

smaller particles. I.e., we find much higher concentrations in the FT than below with almost three times higher concentration at the highest measurement altitude, compared to the residual layer.

**Table 2.** Number size distribution fit parameters on the 05 October 2022 at the different altitude levels. $N_1$, $N_2$ are number concentrations, $D_1$ and $D_2$ are geometric mean diameters and $\sigma_1$ and $\sigma_2$ are geometric standard deviations of the first and second size distribution modes.

| Altitude [m] | $N_1$[cm$^{-3}$] | $D_1$[nm] | $\sigma_1$ [-] | $N_2$[cm$^{-3}$] | $D_2$[nm] | $\sigma_2$ [-] |
|---|---|---|---|---|---|---|
| ∼70 | 44 | 25.9 | 1.79 | 38 | 99.7 | 1.48 |
| ∼150–600 | 68 | 22.2 | 1.93 | 34 | 94.9 | 1.50 |
| ∼1000 | 182 | 18.3 | 2.32 | 26 | 100.4 | 1.56 |
| ∼1500 | 259 | 15.7 | 1.99 | 37 | 99.2 | 1.42 |
| ∼2400 | 344 | 29.9 | 1.94 | 38 | 107.8 | 1.36 |

Additionally to the the number concentration, the SMPS aerosol number size distribution was also investigated at the differ-
ent atmospheric layers (Fig. 11). The measured SMPS scans (1-3) were averaged within the individual layers and are shown as





black dashed lines (please note the different y-axis for the different panels). To be able to follow better the change in the size distribution modes, a double lognormal function was fitted to the measured data and is shown as grey lines. The obtained fit parameters are shown in Table 2.

At the highest altitude, in the FT, we see a sudden increase of the particle number concentration at the lowest size bins (∼10 nm), which can be an indication for new particle formation. However, we only see a small fraction of this mode due to the diameter limit of the SMPS. Therefore, we neither apply a fit to this mode nor speculate about it.

Furthermore, it seems that the increasing aerosol number concentration towards the FT is connected to the increasing number of particles below 40 nm. The concentration of the mode around 90–100 nm seems to stay relatively stable with concentrations between 25–40 cm$^{-3}$, whereas the concentration of the first mode around 15-30 nm particle size increases continuously from

∼40 cm$^{-3}$ in the ABL to more than 300 cm$^{-3}$ in the FT.

The increased fraction of larger particles in the ABL can also be seen here, the number size distribution at diameters larger than the second mode does not converge to zero (better visible with logarithmic y-axis, not shown here), which might indicate the presence of another mode with even larger particle diameters around (400-600 nm). This points also to the presence of sea-spray over the open water emitting larger sea-salt particles into the ABL.

Last but not least, the behaviour of the measured BC mass concentration through the different atmospheric layers will be discussed (Figure 10, black dots with error bars showing the median and 25$^{th}$ and 75$^{th}$ percentiles), as measured by the SP2. The median values ranged between 0.32 ng m$^{-3}$ and 1.83 ng m$^{-3}$. These values are extremely low compared to the median summer (outside of the Arctic haze season) BC concentration of 4.7 ng m$^{-3}$ measured during 2 aircraft campaigns in the European and Canadian Arctic region (Jurányi et al., 2023). The BC mass concentration follows the same pattern as the total

aerosol number concentration with higher values in the FT, and lower ones in the residual and boundary layers. This indicates that the BC is internally mixed and that there is no larger source in the lower atmosphere on Svalbard.

To summarize the overall situation during the case study on 05 October 2022, we have encountered much higher aerosol number and BC mass concentration in the FT than in the layers below. This shows us that the FT is an important aerosol source, and particles are mainly transported downwards to the lower atmospheric layers. The origin of these particles might

be long-range transport and recent new particle formation. The observed increased BC mass is an indication for the former, the presence of the high concentration of small particles for the latter. Next to this, the higher concentration of larger particles close to the open water surface suggests the presence of an additional local aerosol source of sea-spray.

## 4 Conclusions

The T-Bird towed instrument platform represents a valuable tool for studying aerosol and turbulence properties in the challeng-

ing low-altitude regions of the polar atmosphere. Initial tests during the BACSAM campaign demonstrated the system's ability to capture key aerosol parameters, even in areas previously inaccessible to standard aircraft due to altitude constraints. Despite some technical challenges, the results indicate promising performance for future studies.



The comparison between the T-Bird's onboard aerosol instruments and the aircraft's standard instrumentation showed strong correlation in particle number concentration data, when sampling the same air-mass, confirming the functionality of the T-Bird system. Additionally, the ability to perform simultaneous two-level measurements offers new opportunities to study the vertical distribution of aerosols in relation to, e.g. turbulent ABL processes.

Future deployments of the T-Bird, especially under different seasonal and geographic conditions, will enhance its utility in providing critical data for understanding aerosol-cloud interactions, long-range transport of pollutants, and Arctic amplification processes. Continued refinement of the system, particularly in addressing instrumental limitations, will further improve its capabilities and reliability as a key resource for polar atmospheric research.

*Data availability.* The master tracks of the individual research flights during BACSAM can be found in PANGAEA database: https://doi.org/ 10.1594/PANGAEA.958870, https://doi.org/10.1594/PANGAEA.958872, https://doi.org/10.1594/PANGAEA.958873, https://doi.org/10.1594/ PANGAEA.958874, https://doi.org/10.1594/PANGAEA.958875, https://doi.org/10.1594/PANGAEA.958876, https://doi.org/10.1594/PANGAEA. 958877, https://doi.org/10.1594/PANGAEA.958878. The aerosol and turbulence data used in this study is available upon request.

*Author contributions.* Z.J. took part in the planning of the T-Bird and in the BACSAM campaign, analyzed, interpreted the data and wrote the paper. C.L. initiated the T-Bird's development, took place in BACSAM and analysed and interpreted the turbulence data. F.S. functioned as campaign PI during the BACSAM campaign, took part in flight planning and instrument operation on the aircraft, as well as in the analysis and interpretation of the aerosol related data. J. H. took part in the flight campaign and analysed the meteorological and turbulence data. E.G. and M.S. designed and built the T-Bird and operated it during BACSAM. P.H. took part in BACSAM and analysed the flight behaviour of the T-Bird. D. K. helped with the T-Bird development and operated the aerosol instruments during BACSAM. B.W. and J.S. took part in BACSAM and operated the instrumentation. A.S. and D.S. helped with the data analysis and interpretation. S.G. designed the T-Bird's aerosol inlet and helped in the campaign preparations. A. H. took part in the planning of the T-Bird, organised BACSAM and acted as PI, took part in the data interpretation. All authors read and edited the manuscript.

*Competing interests.* The contact author has declared that none of the authors has any competing interests.

*Acknowledgements.* We gratefully acknowledge the funding by the Deutsche Forschungsgemeinschaft (DFG, German Research Foundation) – Projektnummer 268020496 – TRR 172, within the Transregional Collaborative Research Center "ArctiC Amplification: Climate Relevant Atmospheric and SurfaCe Processes, and Feedback Mechanisms (AC)3". Next to it, the authors would like to thank all participants of the BACSAM aircraft campaign. The authors acknowledge support by the Open Access Publication Funds of Alfred-Wegener-Institut Helmholtz-Zentrum für Polar- und Meeresforschung.



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
