# Peer review of "The T-Bird – A new aircraft-towed instrument platform to measure aerosol properties and turbulence close to the surface: Introduction to the aerosol measurement system"

_EGUsphere, 2025_

## Author Comment (AC1)

**Answer to Reviewer 1**

We thank the reviewer for his time spending on the review and for positive comments and constructive hints.

The reviewer comments are displayed in italics, our answers in bold. The modifications in the manuscript text are shown in blue.

*The T-bird is a great idea and the ability to make measurements down low where an aircraft cannot operate opens up great possibilities.*

**Thanks, yes, T-bird offers new possibilities for future research.**

*I would think a more apple-to-apple comparison would have been nice. Since the instruments in the T-bird are not large, having those in the aircraft in addition to what they call the standard instruments would have made for a more apt comparison, in my opinion.*

**We agree, but this could not be organized for this first campaign. In the future we will try the inclusion of the T-bird instrumentation in the aircraft. This is mentioned now in the Conclusion Section, where we write:**

To support better intercomparison and validation of observations, future measurement campaigns will also aim to integrate the T-Bird aerosol instrumentation into the aircraft itself, where feasible. Thereby, it will be kept in mind, that the much higher payload and available space on the aircraft allow for operation of more precise (e.g. ultra-high sensitivity spectromenter, UHSAS instead of POPS) instrumentation.

*I think a good use for the T-bird could be to porpoise the platform up and down to generate a profile of the conditions below the aircraft*

**Yes, this is to some extent also our plan for the future. At least we could obtain wind and temperature profiles. However, for turbulence measurements one always needs longer measurement time in one height for statistical reasons. Also, for the aerosol measurements one might need longer measurement than the nominal 1 Hz time resolution of the Partector 2 and POPS instruments due to the extreme low concentrations. However, the time ~15-minutes (was wrongly stated ~10-minutes in the manuscript, now corrected) that we have spent at a single altitude during most of the flights during BACSAM was not chosen due to the T-Bird instrumentation, it was chosen because of the scanning mobility particle sizer's time resolution (5-minutes) onboard the aircraft. The following texts were added to section 3.3.1:**

The length of the legs was chosen such, that independent of SMPS's exact scan start time, it was guaranteed, that two full scans could be performed at that single altitude.

**And:**

The T-bird also offers the possibility of measuring vertical profiles of meteorological properties such as temperature, pressure, wind components during climb and descent. POPS and Partector have a time resolution of 1Hz, however the Partector has a 4-32 s averaging window,

which is manually set dependent on the expected concentration. For the derivation of turbulence properties (e.g. fluxes, turbulent kinetic energy) from high resolution wind and temperature measurements we need leg lengths at constant altitude of at least 8-10 km for statistical reasons.

*I think I would adjust the title of the paper. There is very little in here about the wind and turbulence part of the platform instrument suite. I am not sure referencing it need to be in the title. If it does, then it shouldn't be listed before aerosol properties which is what the paper is about. I will be interested in seeing the follow on to this paper about the winds and turbulence data. For the little that it talks about it, I do have some questions.*

**Yes, indeed the paper concentrates mainly on aerosol. The name T-Bird reflects the development of this tool during recent years, which concentrated first on turbulence equipment. In the revised version we changed the order in the first line so that the new title is:**

The T-Bird – A new aircraft-towed instrument platform to measure aerosol properties and turbulence close to the surface: Introduction to the aerosol measurement system.

*What Aventech 5-hole probe model are you using. They list nothing on their website that has a response at 100 Hz. Are you using their pressure transducers? Is this an AIMMS instrument. Definitely need more info here and description. At best their stuff updates at 40 Hz and even then their frequency response tails off when you get to 10 hz or so from my experience.*

**Our information about the turbulence equipment was misleading. We do not use an AIMMS20. Only the 5-hole measurement boom is from Aventech including also a purge system for water in tubes after flights in clouds. The latter is used, however, not during measurement legs. So, the boom itself is not involved in data conversion or data storing. We use pressure transducers from Setra (Setra 239 R and Setra 278 , more details in Ehrlich et al., 2019) in combination with NI Compact Rio AD conversion modules reaching a data rate of 100 Hz. This information is given now in the revised version. In some earlier campaigns we used also the AIMMS20, which was installed at the aircraft wing in addition. Thus, we know that our system at the noseboom is faster than the AIMMS20. The following changes were made in the text, in section 2.1.1 (description of the T-Bird turbulence equipment):**

Namely, the same five-hole probe (Aventech) is installed at the front of the T-Bird (Fig. 2) and provides static and dynamic (Pitot) pressure and differential pressure measurements to derive the angles of attack and sideslip and finally the 3D wind vector with a frequency of 100 Hz (Hartmann et al., 2018).

**was changed to:**

Namely, the same five-hole probe (Aventech) is installed at the front of the T-Bird (Fig. 2), wherein Setra 239 R and Setra 278 pressure transducers in combination with NI Compact Rio AD conversion modules provide static and dynamic (Pitot) pressure and differential pressure

measurements to derive the angles of attack and sideslipe and finally the 3D wind vector with a frequency of 100 Hz (Hartmann et al., 2018).

**and in section 2.3 (description of the Polar 6 turbulence instrumentation):**

The aircraft was equipped with a nose boom hosting the instrumentation bunch for turbulence observations. This consist of a five-hole probe (Aventech) and Pt100 temperature sensor (Rosemount, type 102E).

**Was changed to:**

As already mentioned, the aircraft was equipped with a nose boom hosting the instrumentation bunch for turbulence observations. This consist of a five-hole probe (Aventech) with Setra 239 R and Setra 278 pressure transducers and Pt100 temperature sensor (Rosemount, type 102E).

*I haven't seen any other listing of a Rosemount 102 deiced TAT sensor that updates at 100 hz, can you provide some more info. Even if you want to sample that fast, it generally only responds at most at a couple of hertz and much slower than the non-deiced version.*

**Both in the aircraft and in the T-Bird, the temperature sensors are not de-iced. We think that this was a misunderstanding. In the manuscript we only state, that additionally to the 100 Hz not deiced sensors, in the aircraft we have a de-iced temperature sensor mounted as well, which was not used for the data analysis in this manuscript. And as you say, that sensor does not reach a 100 Hz resolution.**

**The text:**

Additionally, there are a deiced (heated) Pt100-temperature sensor (Rosemount, type 102E) and a humidity sensor (Vaisala HMT333) mounted on Polar 6 with dedicated inlets.

**Was changed to:**

Additionally, there are a deiced (heated) Pt100-temperature sensor (Rosemount, type 102E , not used here for data analysis) and a humidity sensor (Vaisala HMT333) mounted on Polar 6 with dedicated inlets.

*All in all I think this is a good introduction to the T-bird but some more details would be nice to see. I am also not an aerosol person so they may have way more to say than I since this turned out to be an aerosol paper. I don't see any showstoppers here but some more fleshed out info would be nice.*

**The main purpose of the manuscript is to introduce the T-Bird as a useful device for characterizing the lowermost layer of the Arctic atmosphere. The data presented here, are therefore mainly to prove the T-Bird's functionality, and outline options for further improvements as well as for future applications. In our opinion, the presented information and data suffice to fulfill that goal. More detailed information will be presented, e.g., in a**

**follow-up paper focusing on the determination of aerosol particle fluxes in the Arctic based on the second scientific T-Bird campaign (BACSAM II).**

References

Ehrlich, A., Wendisch, M., Lüpkes, C., Buschmann, M., Bozem, H., Chechin, D., ... & Zanatta, M. (2019). A comprehensive in situ and remote sensing data set from the Arctic CLoud Observations Using airborne measurements during polar Day (ACLOUD) campaign. *Earth System Science Data*, *11*(4), 1853-1881.

---

## Author Comment (AC2)

**Answers to Reviewer 2**

The authors would like to thank Joshua Schwarz for his generally positive judgement and the constructive comments.

The reviewer comments are displayed in italics, our answers in bold. The modifications in the manuscript text are shown in blue.

*1) "nest" was not described, please add a sentence*

**A sentence describing the nest was added to section 2.1:**

When the T-Bird is found in a winched-in position, it is mechanically fixed in the so-called nest, which is a frame construction located at the bottom of the aircraft fuselage.

*2) please provide temperature control range/ stability in T-BIRD when used, and range as tested/flown.*

**The temperature in the T-Bird was not actively controlled during BACSAM, neither a heating nor a cooling system was installed. The higher temperature inside the T-Bird compared to the ambient temperature was solely a result of the "waste" heat of the inside instrumentation (pumps, electronics). During BACSAM the outside temperature range during the flights varied between -21,4 °C and +5,0°C, which resulted in the internal temperature measured inside the Partector instrument within the range -4,6 and 13,8 °C. The temperature measured in the POPS was always much higher around +10°C due to its internal laser heating. At these encountered temperatures, the instruments in the T-Bird worked properly, however, as it is mentioned in the text, it is planned for future campaigns with even lower outside temperatures to install extra heating in the T-Bird. The text concerning the temperature range during BACSAM in section 2.1.3. now reads:**

The amount of heat produced by the instruments during operation was high enough during the Arctic autumn test campaign BACSAM (minimal measured outside temperature during flight of -21.4 °C) to keep the temperature high enough for the instruments installed inside the T-Bird to work properly. The temperature was not actively controlled, no extra heater nor cooler was installed in the T-Bird. The higher inside temperature of the T-Bird was also enough to keep the relative humidity of the aerosol sample below 40%, no additional drying was used. An optional heating system can be installed for campaigns performed at even lower temperatures present.

*3) I suggest that you cut data/discussion of POPS/SMPS comparison when POPS was not within operational parameters; this does not reflect a properly operating instrument, and so is not relevant.*

Thanks for the suggestion. We have decided to show the size distribution comparison for only the data when the flow was within the measurement range, as suggested. However, for the concentration comparison we would like to keep and show all data. The reason is that the

counting performance is the same, even if that was not expected, and with that we do have better statistics.

*4)Was the SMPS geometric diameter for Partector comparison only calculated from 10-300 nm diameter range? Please specify*

**No, actually, the SMPS geometric diameter was calculated using its full diameter range up to 850 nm. It is true, correct would be to match the nominal diameter range of the Partector and use only the 10-300 nm range. However, due to the shape of the size distribution, the number fraction of the particles above 300 nm is negligible, and therefore it does not really matter (average difference in the geometric diameter of 0.8 nm) if you use the complete SMPS range up to 850 nm or only up to 300 nm. Anyway, the plots were updated with the recalculated data considering the SMPS data only up to 300 nm. The data description now reads:**

The average diameter obtained by the Partector can also be compared to the measurements of the SMPS, and for this comparison the geometric mean diameter of the size distribution was chosen considering the diameter range between 10 and 300 nm.

*5) please address apparent inconsistency between POPS/SMPS concentrations > 300 nm in figure 6b*

**We have already tried to address the inconsistency between the POPS/SMPS sizing, where it seems that the POPS overestimates particles below and underestimates above approx. 250 nm. One possible reason is the sheath flow problem the other is, that we compare a size distribution based on mobility diameter (SMPS) to another, based on optical diameter (POPS). We have no means to decide which effect how much influenced our measurements, therefore we decided not to speculate further about it. A thorough comparison between POPS and SMPS can follow in a next publication (in preparation) based on the follow-up campaign of BACSAM, where the flow problem of POPS was eliminated. We have changed the following text:**

The SMPS measures the number size distribution based on the mobility diameter whereas the POPS measures based on the optical diameter using Polystyrene Latex particles with a refractive index of 1.615+0.001i (Gao et al., 2016) for calibration. However, we expect that this effect is negligible compared to the sizing uncertainty caused by the mis-adjusted sheath flow. The too low, completely missing or even reversed sheath flow has the consequence that aerosol particles that were supposed to pass through the middle of the laser beam might have passed the laser closer to the edge of the beam with significantly lower intensity and therefore falsely identified as smaller particles. In the measured number size distribution, this would appear as measuring too many smaller particles and too few larger ones, just like we have observed in our case.

As the measured number concentration highly correlates to the number concentration of the SMPS, we will still use the POPS data in the following as an indicator for the number concentration of larger (>153 nm) particles.

**To this one, for better understandability:**

The SMPS measures the number size distribution based on the mobility diameter whereas the POPS measures based on the optical diameter using Polystyrene Latex particles with a refractive index of 1.615+0.001i (Gao et al., 2016) for calibration. As atmospheric aerosol particles may feature an in our case unknown size dependent refractive index, a direct qualitative comparison of POPS and SPMS measured size distributions is difficult. Another issue in this context is the too low sheath air flow inside POPS, which has the consequence that aerosol particles that were supposed to pass through the middle of the laser beam might have passed the laser closer to the edge of the beam with significantly lower intensity, and therefore falsely identified as smaller particles. In the measured number size distribution, this would appear as measuring too many smaller particles and too few larger ones, just as our comparison shows. A similar effect can be caused by the presence of size dependent refractive index, if the larger aerosol particles have a significantly lower refractive index than the calibration aerosol. Due to all these uncertainties, in the following we will solely use the POPS data as an indicator for the presence of larger (>153 nm) particles, which is justified due to the good correlation between the POPS's and SMPS's measured number concentration.

*6) Lin e103 - it looks like a calculation error on the total flow into the isokinetic inlet (0.37 cm diameter at 60 m/s->~40 lpm)*

**Yes, the reviewer is correct, thanks a lot. We have made here an error, the total flow is 38.7 lpm. Was corrected in the text.**

7) I suggest combining figure 7 and 8 so that they can be seen together.

**Thanks for the suggestion, the two figures are now combined.**

*8) Line 405 - the BC mass concentrations provide no information about the BC's microphysical mixing state (as suggested by the use of the term "internally mixed"). Perhaps you mean to suggest that the FT airmass is apparently homogeneous?*

**Yes, you are completely right. The BC concentration does not provide information on its mixing state. We actually wanted to suggest that the BC most probably arrives from sources far away (therefore probably would also be internally mixed, but we have of course no information on that). The corrected sentence now reads:**

The BC mass concentration follows the same pattern as the total aerosol number concentration with higher values in the FT, and lower ones in the residual and boundary layers. This indicates that the BC is not freshly emitted and there is no larger source in the lower atmosphere on Svalbard.

*9) For context: is the instrument payload of TBIRD imagined to be adjustable for different missions, or is it fixed effectively permanently?*

**The payload of the T-Bird is not fixed, it is possible to modify it and place other instruments inside. However, we have to mention that any modification of the system would require a completely new certification of the system as well, which means that such an action will have to be planned well in advance (1-2 years at least) and requires significant financial and human resources. We have added the following text to the manuscript to section 2.1:**

The T-Bird's instrumental payload could also be adjusted, if required. However, it should be kept in mind, that changing the T-Bird's configuration require a new certification, which has to be planned well in advance and requires both, human and financial resources.

*10) The section describing the results shown in figure 10 was a bit confusing, and would benefit from careful editing. For the figure, panel C, I wonder why the 100m TBIRD point is not associated with sampling error bars, and why the associated POLAR-6 point at ~150m appears to show enhanced flux (based on simply looking at Ramanelli and Zardi, AMT 2004 figure 2 to establish expectations…). Is this a cause for concern?*

**Thanks for the comment. The shown error bars in panel c are showing the sampling error. We calculated those following Srenivasan et al. (1978) and Fiedler et al. (2010), due to which it goes to zero when the flux approaches zero and the flight leg is long. This explains that at 100 m the error bar is not seen (smaller than the size of the marker).  Furthermore, one should not overinterpret the results for heat flux. Usually, one would not expect a higher accuracy than plus/minus 2 W m-2 because of further errors other than the sampling error, e.g. caused by inhomogeneity during the leg and intermittent turbulence especially when the aircraft is close to the inversion base (see also Tetzlaff et al., 2015). Nevertheless, in this slightly convective case the linear decrease from a maximum near the surface to very small and sometimes negative (downward) fluxes in the inversion is physically reasonable. We have significantly modified, and hopefully with this improved the description of figure 9 (which the comment meant instead of figure 10).  The description now reads:**

The temperature profile at WP1 (Figure 9) reveals that the structure of the lower atmosphere is characterized by multiple inversions and mixed layers in between. The lowest one is the strong ABL capping inversion at about 100-130 m height. Above the inversion a well-mixed layer (in the following called residual layer) follows again. And also this layer is capped by a strong inversion at about 750 m height, which reaches to about 1000 m}. Then, another layer follows, which is also mixed but not so well as the residual layer. Finally, another inversion starts at about 1500 m height.

The leg averages show the potential temperature, wind speed (Figure 8 a,b) and turbulence quantities (Figure 8 c,d), shea sensible heat flux and turbulent kinetic energy (TKE). For the calculation of these turbulence quantities by the eddy covariance method, the linear trends of wind and temperature between both WPs have been eliminated. Both the altitude dependent potential temperature and heat fluxes (both obtained from the horizontal flight legs, Figure 8 a,c) point to a weak convectively mixed ABL below about 100 m height with small upward fluxes of sensible heat. It can be seen that the results from T-Bird (purple markers) and Polar 6  (green markers) fit very well to the results obtained from Polar 6 (green markers). This concerns especially potential temperature and wind with a weak low level jet and turbulent  kinetic energy. Also the sensible heat fluxes are reasonable but at 150 m, the heat flux seems to be overestimated by Polar 6 since negative values could be expected near the inversion bottom. However, one should not overinterpret the heat flux values, since they are close to the detection limit. Usually, one would not expect an accuracy larger than +-2W m-2 because of further errors other than the sampling error (shown in the figure as error bars), which are calculated following Sreenivasan et al. (1978) and Fiedler et al. (2010). E.g., especially near the inversion bottom, inhomogeneity can occur along the leg as well as intermittent turbulence, which makes the measurements less reliable during such legs (see

Tetzlaff et al., 2015). In such cases the measured flux profile can deviate from its ideal shape described, e.g. in Rampanelli and Zardi (2004). Nevertheless, it is impressing that although the ABL is only slightly convective and heat fluxes are very small in this considered case, the expected linear decrease from a maximum of heat flux near the surface to very small and sometimes negative (downward) values in the capping inversion is reproduced by the measurements. Also, the TKE altitude dependence with a maximum near the surface is physically reasonable.

**And the following text was added to the label of Figure 8 (9 in the old version):**

The sampling error in Panel c at about 100 m is almost zero because it is proportional to the measured flux (Sreenivasan et al., 1978), which is close to zero at this point.

*11) some information about the relative scale of TBIRD motion (other than directly forward) to wind/turbulence measurement would be usefule for context/uncertainty evaluation.*

**The following text was added to the flight behavior section:**

T-Bird's motion relative to the towing plane represents a pendulum movement with a period depending on the length of rope. The largest amplitude of this movement is parallel to the flight direction and is a result of towing force changes due to vertical movement of the plane. This movement shows in speed ondulations of the bird. It has a negligible effect on the accuracy of the wind measurement as this movement is very precisely measured by the inertial system. Pendulum movement across flight direction results from turns of the aircraft to align on a desired track. The across movement eases out after two or three pendulum periods if the aircraft is flying steady on a straight track. The aerodynamics of the bird lead to very low sideslip angles, typically less than one degree during straight measurement flights. Even in regular turns sideslip angles greater than 2 or 3 degrees are very rarely exceeded. Thus the 5-hole-probe is nearly always in its specified and calibrated range.

*12) Can you use data from TBIRD in the climb/descent portions of the flights? If data collection is limited to level legs (even if just for aerosol concentrations/size), that is an important limitation to mention.*

**Pressure, temperature and wind components (derived from nose boom pressure measurements) are continuously measured also during climb/descent. The determination of turbulence properties with high accuracy requires horizontal leg lengths of at least 8-15 km depending on atmospheric stability (sizes of energy transporting eddies). However, if one accepts a reduced accuracy one can derive also turbulent flux profiles (see, Chechin et al., 2023). The following text was added to section 3.3.1.:**

The T-bird also offers the possibility of measuring vertical profiles of meteorological properties such as temperature, pressure, wind components during climb and descent. POPS and Partector have a time resolution of 1Hz, however the Partector has a 4-32 s averaging window, which is manually set dependent on the expected concentration. For the derivation of turbulence properties (e.g. fluxes, turbulent kinetic energy) from high resolution wind and temperature measurements we need leg lengths at constant altitude of at least 8-10 km for statistical reasons.

**References**

Chechin, D. G., Lüpkes, C., Hartmann, J., Ehrlich, A., & Wendisch, M. (2023). Turbulent structure of the Arctic boundary layer in early summer driven by stability, wind shear and cloud-top radiative cooling: ACLOUD airborne observations. *Atmospheric Chemistry and Physics*, *23*(8), 4685-4707.

Fiedler EK, Lachlan-Cope TA, Renfrew IA, King JC. 2010. Convective heat transfer over thin ice-covered coastal polynyas. *J. Geophys. Res.* **115**: C10051, doi: 10.1029/2009JC005797

Sreenivasan KR, Chambers AJ, Antonia RA. 1978. Accuracy of moments of velocity and scalar fluctuations in the atmospheric surface layer. *Boundary-Layer Meteorol.* **14**: 341–359, doi: 10.1007/BF00121044.

Tetzlaff, A., Lüpkes, C., & Hartmann, J. (2015). Aircraft-based observations of atmospheric boundary-layer modification over Arctic leads. *Quarterly Journal of the Royal Meteorological Society*, *141*(692), 2839-2856.